# Design of a Prism-Grating Wide Spectral Range Transmittance Imaging Spectrometer

**DOI:** 10.3390/s23115050

**Published:** 2023-05-25

**Authors:** Xu Zhang, Bo Li, Xue Jiang, Guochao Gu, Hanshuang Li, Xiaoxu Wang, Guanyu Lin

**Affiliations:** 1Changchun Institute of Optics, Fine Mechanics and Physics, Chinese Academy of Sciences, Changchun 130033, China; zhangxv22@mails.ucas.ac.cn (X.Z.); jiangxue470@163.com (X.J.); ggc-2003@163.com (G.G.); lihanshuang06@163.com (H.L.); wangxiaoxu@ciomp.ac.cn (X.W.); linguanyu1976@163.com (G.L.); 2University of Chinese Academy of Sciences, Beijing 101408, China

**Keywords:** imaging spectrometer, wide spectrum band, achromatic, secondary spectrum, miniaturization

## Abstract

As spectroscopic detection technology rapidly advances, back-illuminated InGaAs detectors with a wider spectral range have emerged. Compared to traditional detectors such as HgCdTe, CCD, and CMOS, InGaAs detectors offer a working range of 400–1800 nm and exhibit a quantum efficiency of over 60% in both the visible and near-infrared bands. This is leading to the demand for innovative designs of imaging spectrometers with wider spectral ranges. However, the widening of the spectral range has led to the presence of significant axial chromatic aberration and secondary spectrum in imaging spectrometers. Additionally, there is difficulty in aligning the system optical axis perpendicular to the detector image plane, resulting in increased challenges during post-installation adjustment. Based on chromatic aberration correction theory, this paper presents the design of a wide spectral range transmission prism-grating imaging spectrometer with a working range of 400–1750 nm using Code V. The spectral range of this spectrometer covers both the visible and near-infrared regions, which is beyond the capability of traditional PG spectrometers. In the past, the working spectral range of transmission-type PG imaging spectrometers has been limited to 400–1000 nm. This study’s proposed chromatic aberration correction process involves selecting optical glass materials that match the design requirements and correcting the axial chromatic aberration and secondary spectrum, ensuring that the system axis is perpendicular to the detector plane and easy to adjust during installation. The results show that the spectrometer has a spectral resolution of 5 nm, a root-mean-square spot diagram less than 8 μm over the full field of view, and an optical transfer function MTF greater than 0.6 at a Nyquist frequency of 30 lp/mm. The system size is less than 90 mm. Spherical lenses are employed in the system design to reduce manufacturing costs and complexity while meeting the requirements of wide spectral range, miniaturization, and easy installation.

## 1. Introduction

Imaging spectrometers, as optical remote sensing sensors, enable the acquisition of both spatial and spectral information. In recent years, they have found extensive applications in various fields, including land, ocean, and atmosphere [1]. For instance, they have been utilized for space atmospheric detection, Earth’s water pollution detection, geological observation, Earth’s environmental status monitoring, crop damage detection, and more. In space atmospheric detection, imaging spectrometers can detect the concentration and distribution of trace gases, contributing to a better understanding of atmospheric composition and the impact of human activities on the environment. In Earth’s water pollution detection, imaging spectrometers can detect and monitor the concentration of pollutants in water, providing vital information for water quality monitoring and management. The advancement of imaging spectrometer technology has led to an increased demand for high resolution, wide spectral coverage, and miniaturization of imaging spectrometers. Wide spectral coverage enables the detection and acquisition of more object information, facilitating object identification and classification [2]. The miniaturization of imaging spectrometers can overcome platform type constraints, thereby enabling deep space exploration and meeting the needs of civil platform use. The advantages of low processing cost and a wider application scope further enhance their attractiveness for diverse applications.

Currently, the common structure of imaging spectrometers is mostly off-axis, which makes the system difficult to assemble and adjust. In addition, the processing of concave and convex reflection gratings is difficult and expensive, which is not conducive to the deployment of these instruments in the civilian market. As a result, the prism-grating [3] type imaging spectrometer has become an important research direction due to its relatively simple components and low processing cost. It possesses the following characteristics, including direct viewing capability. The entire system employs transmissive devices, ensuring co-axial alignment of all optical elements starting from the slit. By selecting the center wavelength and grating constant, a prism is designed to achieve unbiased deflection imaging of the center wavelength. This is highly advantageous for the overall mechanical structure and alignment of the system. Another significant advantage of the prism-grating type imaging spectrometer is its ability to eliminate smile. When the prism and grating are used separately as dispersive elements, they each have spectral curvature, with opposite curvature directions. However, combining the two in a prism-grating system can effectively eliminate smile [4]. The total dispersion effect of the system is approximately equal to the sum of the grating and prism dispersion effects. However, the grating plays a more significant role in dispersion due to its higher efficiency compared to the prism. The compactness of the prism-grating type imaging spectrometer is also a significant advantage, as the optical components are tightly connected, making it conducive to miniaturization and lightweight design. Miniaturization is a key research direction in spectral technology due to its potential for reducing the size and cost of the system, as well as increasing portability. However, the prism-grating type imaging spectrometer also has axial chromatic aberration and secondary spectrum, which can affect the imaging quality. Traditionally, tilting the image plane has been used to improve imaging quality, but this can be detrimental to later adjustments. Therefore, effective methods must be used to eliminate the axial chromatic aberration and secondary spectrum of the prism-grating type imaging spectrometer system [5]. In summary, the prism-grating type imaging spectrometer offers several key advantages over traditional off-axis spectrometers, including direct viewing capability, elimination of smile, compactness, and potential for miniaturization. While challenges such as axial chromatic aberration and secondary spectrum must be addressed, the prism-grating type imaging spectrometer holds promise for a wide range of applications in spectral technology.

In response to the chromatic aberration issue in wide-spectral-range prism-grating imaging spectrometers, this paper designs a prism-grating combination spectral splitting element-based transmissive wide-spectral-range imaging spectrometer, the working range of which is from 400 to 1750 nm, based on the principle of wide-spectral-range chromatic aberration correction. By selecting achromatic lens materials and solving for the focal power of different materials, the obtained focal power is reasonably combined to eliminate the axial chromatic aberration and secondary spectrum of the system. The prism-grating spectrometer adopts a structure in which both the grating and prism are used to separate the spectrum, which not only corrects the smile but also enhances the dispersion performance [6]. The forward telescope system and the spectral imaging system are designed independently, which have the advantages of strong versatility and separate assembly without interference. The wide-spectral-transmission prism-grating imaging spectrometer designed in this paper has a wider working spectral range and can detect the spectrum of 400–1750 nm in one channel, with the advantages of miniaturization and compactness, compared with the traditional PG imaging spectrometer.

## 2. Principle and Design Process

### 2.1. Wide-Spectral-Band Achromatic Reduction Principle

The widening of the working spectral range of the imaging spectrometer has led to serious axial chromatic aberration and secondary spectrum in the system, which affects the imaging quality of the system [7]. As shown in Figure 1, tilting the image plane can improve the imaging quality, but this structure is not conducive to later installation and adjustment, and there is axial chromatic aberration that causes spectral crosstalk during detection. To solve the problems of axial chromatic aberration, secondary spectrum, and image plane tilting in the wide spectral range transmission prism-grating spectrometer designed in this paper, we use the principle of chromatic aberration correction and the selection of chromatic aberration correction materials to eliminate the axial chromatic aberration and secondary spectrum of the prism-grating imaging spectrometer, while ensuring that the optical axis is perpendicular to the image plane, so that it has good imaging quality in the working spectral range of 400~1750 nm.

It can be observed from the Sellmeier formula [8], which is a standard formula used in optical engineering, that the relationship between a material’s refractive index and the incident wavelength is described by Equation (1):(1)$$n^{2}=1+\frac{B_{1}\lambda^{2}}{\lambda^{2}-C_{1}}+\frac{B_{2}\lambda^{2}}{\lambda^{2}-C_{2}}+\frac{B_{3}\lambda^{2}}{\lambda^{2}-C_{3}}$$

In the above equation, B1–B3 and C1–C3 are given constants, and λ is the incident light wavelength.

Equation (1) demonstrates that the refractive index of glass decreases as the wavelength increases, resulting in a smaller angle of refraction for light passing through the glass. Therefore, when light of different wavelengths enters the same glass material, there is a difference in the imaging position, which leads to chromatic aberration. Chromatic aberration can be categorized into axial chromatic aberration and lateral chromatic aberration [9], as illustrated in Figure 2. Axial chromatic aberration refers to the difference in the imaging position of light of different wavelengths in the axial direction, while transverse chromatic aberration refers to the difference in the imaging height of light of different wavelengths on the image plane [10]. The magnitudes of axial and transverse chromatic aberration are given by Equations (2) and (3), respectively. For the prism-grating imaging spectrometer used in this design, the working spectral range is 400~1750 nm, with short, center, and long wavelengths represented by $\lambda_{1}$ = 400 nm, $\lambda_{2}$ = 1075 nm, and $\lambda_{3}$ = 1750 nm.

Axial chromatic aberration expression:(2)$$\Delta l^{\prime }{}_{\lambda_{1}\lambda_{3}}=l^{\prime }{}_{\lambda 1}-l^{\prime }{}_{\lambda 3}$$

Vertical chromatic aberration expression:(3)$$\Delta y^{\prime }{}_{\lambda_{1}\lambda_{3}}=y^{\prime }{}_{\lambda_{1}}-y^{\prime }{}_{\lambda_{3}}$$

According to the Literature, the dispersion characteristics of materials are primarily determined by the material’s Abbe number and relative dispersion parameter. Therefore, to investigate the dispersion characteristics of materials, it is necessary to introduce the concepts of Abbe number and relative dispersion parameter [11]. Their expressions are presented below.
(4)$$\begin{array}{l} \upsilon =\frac{n_{\lambda_{2}}-1}{n_{\lambda_{1}}-n_{\lambda_{3}}}\\ p=\frac{n_{\lambda_{2}}-n_{\lambda_{3}}}{n_{\lambda_{1}}-n_{\lambda_{3}}} \end{array}$$

In the equation above, “υ” represents the Abbe number, “p” represents the relative dispersion parameter, and “$n_{\lambda_{1}},n_{\lambda_{2}},n_{\lambda_{3}}$” represents the refractive index of the material for different wavelengths. According to the primary aberration theory [12], a condition for eliminating axial chromatic aberration in an optical system is given by:(5)$$C_{1}=\sum_{i=1}^{N}\frac{\Phi_{i}}{\upsilon_{i}}=0$$

In the above equation, “$\Phi_{i}$” represents the focal power of a lens in an imaging spectrometer system, “$\upsilon_{i}$” represents the Abbe number of the lens material, and “N” represents the number of lenses in the system. Assume that the relationship between the combined lens focal length and the total system focal length in the imaging spectrometer system under the condition of a closely spaced thin lens group is shown by Formulas (6) and (7) [13].
(6)$$\sum_{i=1}^{N}\Phi_{i}=\Phi$$
(7)$$\Phi =\frac{1}{f^{\prime }}$$

In the above equation, “Φ” represents the total focal power of the imaging spectrometer system, and “$f^{\prime }$” represents the focal length of the system. By using Equations (4)–(7), the numerical distribution of the focal power for different materials used for chromatic aberration correction can be calculated. Even with appropriate selection of chromatic aberration correction materials and combinations of positive and negative lenses to correct for axial chromatic aberration between short and long waves, the image point for the center wavelength still does not coincide with those of the short and long wavelengths. This residual chromatic aberration is known as secondary spectrum, which persists in the system after correction of axial chromatic aberration.

The expression for the secondary spectrum is shown in Equation (8):(8)$$\Delta L^{\prime }{}_{\lambda }=mf^{\prime }$$

In the above formula, “m” represents the secondary spectral coefficient, and “$f^{\prime }$” represents the system focal length, which is determined by the material used in the optical system.
(9)$$m=\frac{p_{1}-p_{2}}{\upsilon_{1}-\upsilon_{2}}$$

Therefore, the elimination of the secondary spectrum in the system must satisfy the equation derived from Equations (8) and (9), as follows.
(10)$$\Delta L^{\prime }{}_{\lambda }=\frac{p_{1}-p_{2}}{\upsilon_{1}-\upsilon_{2}}f^{\prime }=0$$

For common optical materials, when the system operates in the visible light range of 400–800 nm, the secondary spectrum coefficient m≈0.00052 can be calculated from Equation (9). The prism-grating imaging spectrometer designed in this paper operates in the visible to near-infrared range of 400–1750 nm, and m≈0.0046 is deduced accordingly. It can be inferred that as the operating spectral range widens, the secondary spectrum coefficient increases and the system chromatic aberration increases accordingly.

Based on Equations (5)–(7), and (10), it is evident that the Abbe numbers of the chosen glass materials should differ considerably, while the relative dispersion coefficients should have only minor differences to minimize the secondary spectral coefficient and eliminate the secondary spectrum within the system. Therefore, it is possible to correct the axial chromatic aberration and secondary spectrum of the transmissive optical system by selecting appropriate materials with low chromatic aberration and distributing the optical focal power of the combined lenses reasonably.

### 2.2. Methods and Processes

This article proposes a method for eliminating axial chromatic aberration and secondary spectrum in prism-grating imaging spectrometers based on theoretical deduction. The flowchart of the system for eliminating axial chromatic aberration and secondary spectrum in the prism-grating imaging spectrometer is illustrated in Figure 3. The following is the design of a transmissive prism-grating imaging spectrometer with a working spectral range of 400~1750 nm, based on the chromatic aberration correction method and process established using the wide spectral range chromatic aberration correction principle.

To start, choose optical glass materials from CDGM or SCHOTT libraries [14] that can operate in the 400–1750 nm wavelength range, while ensuring they meet requirements for acid and corrosion resistance, low coefficient of thermal expansion, and high thermal conductivity. Following a thorough screening process, 14 types of glass materials that meet the design specifications are obtained. Table 1 displays the refractive index, Abbe number, and relative dispersion parameters of these materials within the 400–1750 nm wavelength range.

The designed focal length of the telescope system discussed in this paper is 147 mm, with a spatial magnification ratio of 1:1 for the spectrometer. Therefore, the focal length of the spectrometer system is determined to be $f^{\prime }=147\mathrm{mm}$. As mentioned in the previous section, the chosen achromatic material should have a large Abbe number difference and a small relative dispersion coefficient. Table 1 has been screened to identify a material combination that satisfies these conditions. The secondary spectrum values for each material combination have been calculated using Formula (10), and the results are presented in Table 2. The achromatic material for this study has been selected based on the material combination with the lowest secondary spectrum value.

Table 2 provides a means to determine the optimal combination of N-PK52A and HZF6 materials, which minimizes the secondary spectrum of the system while also satisfying the tolerable focus color shift. By utilizing Equations (5)–(7), the optical power of lenses made from these materials can be determined, with values of $\Phi_{1}=0.01$ and $\Phi_{2}=-0.0038$ assigned to each, respectively. Utilizing only two lenses in the system will not correct for all aberrations. Therefore, by dividing and combining the optical powers of each lens type, the system can correct for additional aberrations, while simultaneously increasing the number of optimizable parameters.

A design process has been developed to eliminate the axial chromatic aberration and secondary spectrum of prism-grating or prism-grating-prism imaging spectrometer systems. Firstly, optical glass materials meeting the working wavelength range and appropriate chemical and physical properties are searched for in the optical glass material library. The Abbe number and relative dispersion coefficient of the selected glass materials are then calculated. According to the chromatic aberration correction principle, materials with large differences in Abbe numbers and relatively small relative dispersion coefficients are selected. Secondly, the secondary spectrum of the selected material combination is calculated to obtain the combination with the smallest secondary spectrum. Finally, the optical power corresponding to each lens of the material combination is calculated, and the remaining aberrations of the system are corrected by segmenting and combining the optical power. Through this design process, the axial chromatic aberration and secondary spectrum are effectively eliminated. As a result, the prism-grating imaging spectrometer achieves excellent imaging quality, ensuring perpendicularity between the image plane and the optical axis. This facilitates the system’s installation and adjustment during later stages.

## 3. Optimization Design of Optical Systems

### 3.1. Design Metrics

The prism-grating imaging spectrometer design presented in this article operates in the spectral range of 400–1750 nm. The Sony COTS-IMX990 InGaAs model is selected as the detector, which features an individual pixel size of 5 μm × 5 μm and a total of 1280 × 1024 pixels. The effective detector area is 6.4 × 5.12 mm, thereby limiting the spectral dispersion length to 5.12 mm. The spectrometer employs a grating with a line density of 141 g/mm and uses +1 order diffraction light, with a ZnS substrate. Table 3 presents the main technical specifications of the system.

### 3.2. Design of a Forward Telescope System

The prism-grating imaging spectrometer comprises various components including a forward telescope system, a slit, a spectrograph, and a detector. The object image is formed onto the slit through the forward telescope system, which is then connected to the spectrograph. Based on the system design specifications and performance requirements presented in Table 3, a co-axial Cassegrain configuration [15] has been selected for the forward telescope system, as shown in Figure 4. This reflective structure offers the benefits of achromaticity and a foldable optical path, reducing the system length and making it suitable for a wide spectral range. The coaxial double reflection system, combined with a correcting lens, expands the field of view and improves the aberration correction capability of the system. This design meets the parameter requirements of the forward telescope and has a simple and compact structure, ensuring that the image is focused at the far point and effectively eliminating the vignetting at the edge of the field of view. The forward telescope system has a focal length of 147 mm, an F# of 3.34, and a system size of 61 × 48 × 48 mm. Positive and negative lenses are utilized to correct system aberrations in the forward telescope system.

This article employs Code V for optimizing the optical configuration of the telescope system. The optimized MTF curve and system spot diagram are presented in Figure 5 and Figure 6, respectively. It can be observed from the figures that the MTF of the system is above 0.7 at the Nyquist frequency of 40 lp/mm for each field and each band. Moreover, the RMS of the spot diagram under the conditions of full field and full band is below 5.5 μm, which satisfies the requisite standards for practical applications.

### 3.3. PG Spectrophotometer System Design

#### 3.3.1. Prism-Grating Spectral Separation Principle

The PG component achieves spectral separation by gluing an ordinary prism and a Volume Bragg grating. The entire component is shown in Figure 7. Parallel light undergoes dispersion by the prism, it is dispersed at different angles depending on the wavelength, and then enters the Volume Bragg grating for further diffraction. The substrate material of the Volume Bragg grating is different from that of the prism. Light of different wavelengths dispersed by the prism is refracted by the substrate into the grating for further diffraction. By adjusting the apex angle of the prism and the groove density of the grating, the central wavelength of the Bragg diffraction can be ensured to be deflection-free along the optical axis, achieving co-axial direct-view design.

#### 3.3.2. Diffraction Efficiency of Volume Bragg Grating

The diffraction efficiency of PG mainly refers to the diffraction efficiency of Volume Bragg gratings. Therefore, the calculation of diffraction efficiency can be analyzed and computed using rigorous coupled-wave analysis (RCWA) [16], scalar coupled-wave theory [17], matrix-based methods, and multiple-beam interference theory [18]. The rigorous coupled-wave theory can obtain accurate theoretical diffraction efficiency, but the equation must be solved using numerical methods, which can be quite complex. The multi-beam interference theory is an approximate discussion method. The results obtained from scalar coupling are shown in Equations (11)–(13):(11)η=sin2υ2+ξ21+ξ2υ2
(12)$$\upsilon =\frac{\pi \Delta nh}{\lambda \cos\theta_{0}}$$
(13)$$\xi =\Delta \theta \frac{\pi h}{\Lambda }$$

In the above equation, h represents the thickness of the grating recording medium, which is dichromate gelatin (DCG) or photo-sensitive polymer deposited on the glass substrate of the grating. The symbol Λ represents the period of the grating. $\theta_{0}$ represents the incident angle of the Bragg wavelength on the incident grating, Δθ represents the deviation of the Bragg incident angle, and Δn represents the degree of refractive index modulation. When only the first-order spectrum is used and in the case of non-polarization, the diffraction efficiency can be expressed by Equation (14):(14)$$\eta_{1}=\frac{1}{2}\left.\left\{\sin^{2}\left(\frac{\pi \Delta nh}{\lambda \cos\theta }\right)\right.+\sin^{2}\left[\frac{\pi \Delta nh}{\lambda \cos\theta }\cos\left(2\theta_{\alpha }\right)\right]\right\}$$

The symbol $\theta_{\alpha }$ represents the angle between the incident and diffracted light rays. Equation (14) shows that the diffraction efficiency of this grating depends on the thickness of the recording medium, the grating period, and the incident wavelength. Based on Equation (14), the incident and diffraction angles of each wavelength band on the holographic grating can be calculated using MATLAB, and the relationship between the diffraction efficiency of the Volume Bragg gratings and the thickness and wavelength of the recording medium can be obtained. However, considering the wide band of the entire system, it is not possible to choose the wavelength corresponding to the highest diffraction efficiency as the center wavelength, because the rate of decrease in diffraction efficiency in the short wavelength direction is much larger than that in the long wavelength direction, relative to the center wavelength. Therefore, the recording medium thickness is selected as 0.022 mm for the entire wavelength band, so that the diffraction efficiency for both the edge wavelength bands and the center wavelength is greater than 0.6. The diffraction efficiency reaches its peak at 1075 nm, reaching about 80%.

#### 3.3.3. PG Imaging Spectrophotometer System Design

The PG-type imaging spectrometer comprises a collimation system, prism-grating, imaging system, and detector. The asymmetrical structure of the collimation and imaging systems increases the degree of freedom to optimize the system and corrects imaging aberrations to improve image quality. Although diffracted light from the prism and grating is deviated in the opposite direction to the incident light, the dispersion direction remains the same [19]. By combining them into a refractive-diffractive hybrid spectrometer element, the deflection angle of the incident and outgoing light of a certain wavelength can be adjusted to zero [20]. This forms a straight-through and compact spectrometer element, as shown in Figure 8.

In order to achieve a co-axial visualization design of the spectrometer system, the design parameters of the grating and prism were determined based on parameters such as spectral range, spectral resolution, system F#, and dispersion length. The prism has two surface tilt angles of 2.9696° each, and is made of ZnS material. The grating has 141 grooves per millimeter and a diffraction order of +1.

The combination of lens material and optical power that eliminates axial chromatic aberration and secondary spectrum in the spectrometer, as well as the design parameters presented in Table 3, were determined through analysis in the previous section. The spectrometer was designed and optimized using Code V, resulting in the achromatic design depicted in Figure 9. The achromatic structure measures 84 × 14 × 14 mm and has a spatial magnification of 1:1. The collimation system has a focal length of 33.249 mm, while the imaging system has a focal length of 24.7524 mm. Figure 8 illustrates the MTF curve for each field of view and wavelength band of the spectrometer, indicating that the MTF is superior to 0.6@25 lp/mm. Additionally, Figure 10 displays the spot diagram for the entire field of view and wavelength band, with a RMS value less than 8 μm.

### 3.4. System Integration

After integrating the aforementioned forward telescope system with the spectroscopic system, the overall structure is shown in Figure 11. The MTF was used to evaluate the system’s performance across the entire spectral band. MTF curves were calculated for the center and edge wavelengths based on the detector’s pixel size, up to a cutoff frequency of 25 lp/mm, as depicted in Figure 12 and Figure 13. At the center wavelength, the MTF is close to the diffraction limit, reaching 0.8, while at the edge wavelength, the MTF is above 0.7. The RMS of the full-field spot diagram is less than 8 μm over the entire spectral range, and the system’s overall size is 145 × 48 × 48 mm. Our design achieves image-plane perpendicularity and coaxially with direct vision, making it ideal for a prism-grating imaging spectrometer with a broad spectral range of 400–1750 nm.

## 4. Conclusions

This article presents a method and process for eliminating axial chromatic aberration and secondary spectrum in prism-grating imaging spectrometers based on the achromatic reduction principle. First, optical materials suitable for the visible-near-infrared wavelength range were selected, and the refractive index of different materials at different wavelengths was calculated using the standard Sellmeier formula to obtain the Abbe number and relative dispersion parameters of the optical materials. Based on the secondary spectrum formula, optical material combinations with large differences in Abbe numbers and small differences in relative dispersion parameters were selected. Then, the secondary spectrums of different optical material combinations were calculated, and the combination with the smallest secondary spectrum was selected. Finally, the focal power was calculated using the formula and different materials were combined to achieve reasonable structural combinations. The chromatic aberration correction method and process yielded materials that were free of chromatic aberration and suitable for the design of a transmission-type prism-grating imaging spectrometer with a wide spectral range of 400–1750 nm. The spectrometer had an RMS point spread function of less than 8 μm, a minimum MTF value greater than 0.6@25 lp/mm, and a spectral resolution of 5 nm. Compared with traditional PG imaging spectrometers, the design of the wide-spectrum transmission-type prism-grating imaging spectrometer presented in this article has a wider working spectral range, and a single channel can detect the entire spectrum of 400–1750 nm. The design is also more compact and easier to assemble, achieving a compact and miniaturized spectrometer design.

## Figures and Tables

**Figure 1 sensors-23-05050-f001:**
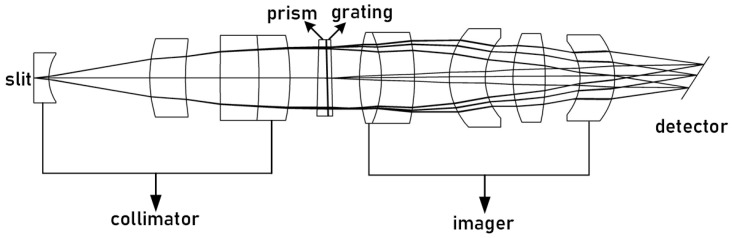
Schematic of spectrometer with tilted image plane.

**Figure 2 sensors-23-05050-f002:**
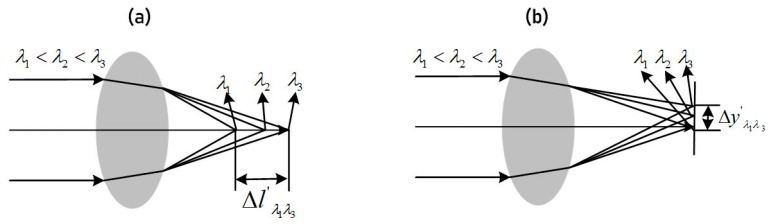
(**a**) Schematic of axial chromatic aberration; (**b**) schematic diagram of lateral chromatic aberration.

**Figure 3 sensors-23-05050-f003:**
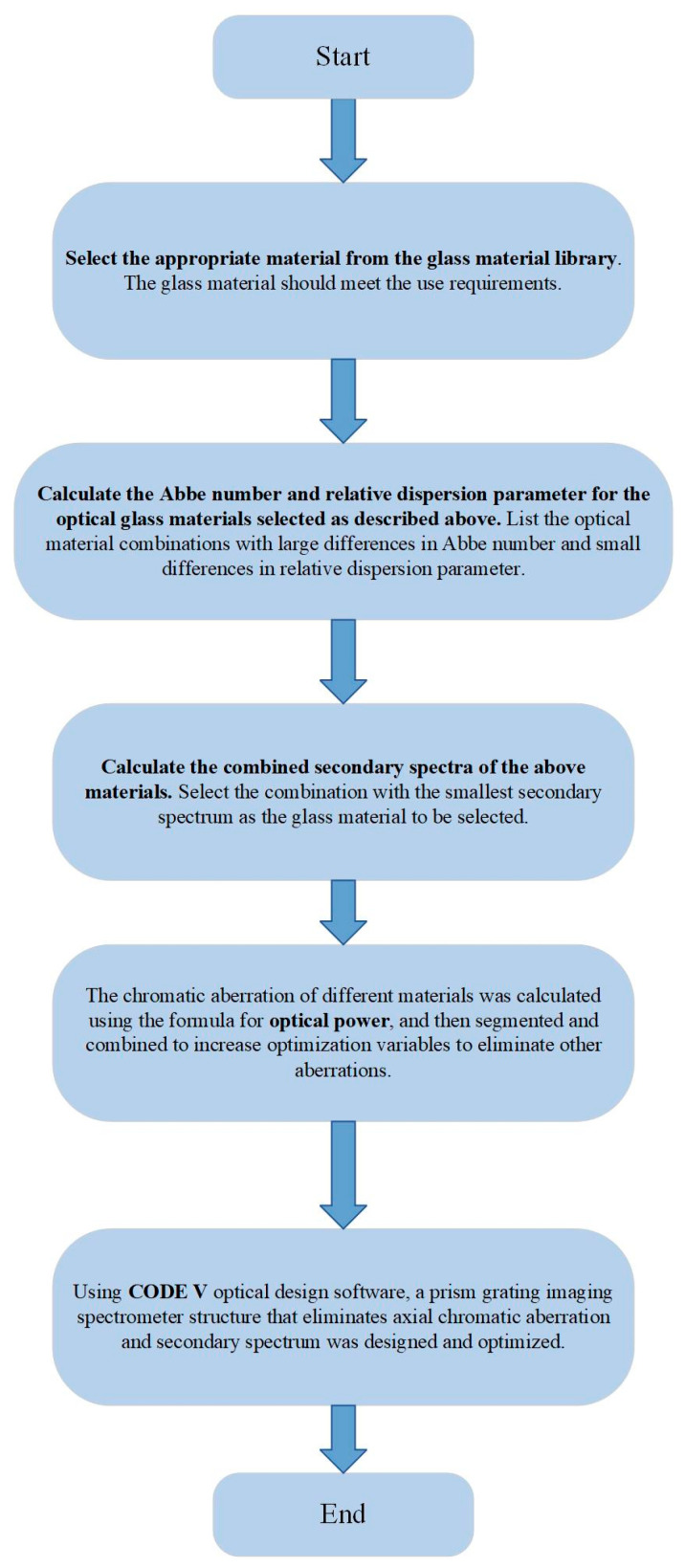
Process flowchart for eliminating axial chromatic aberration and secondary spectra.

**Figure 4 sensors-23-05050-f004:**
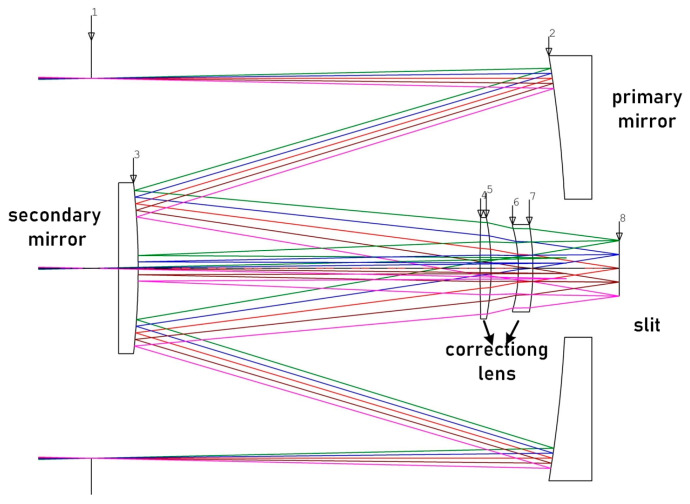
2D view Schematic diagram of the forward telescope system.

**Figure 5 sensors-23-05050-f005:**
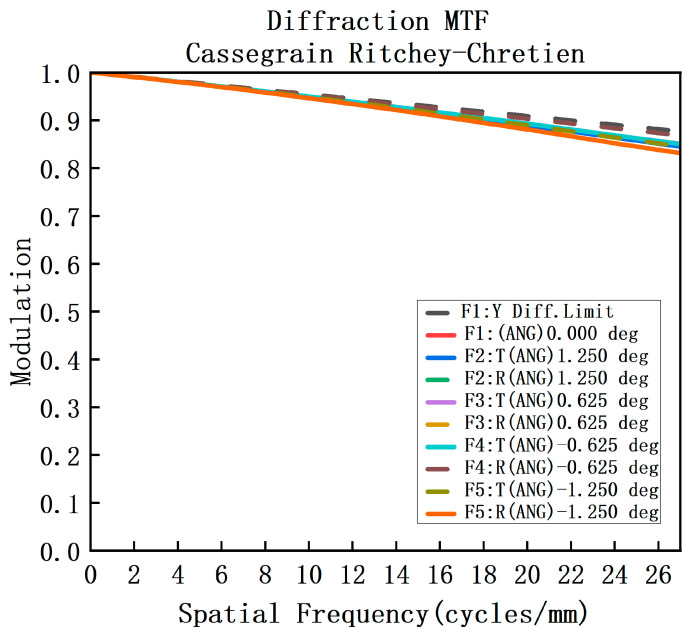
MTF diagram of the forward telescope system.

**Figure 6 sensors-23-05050-f006:**
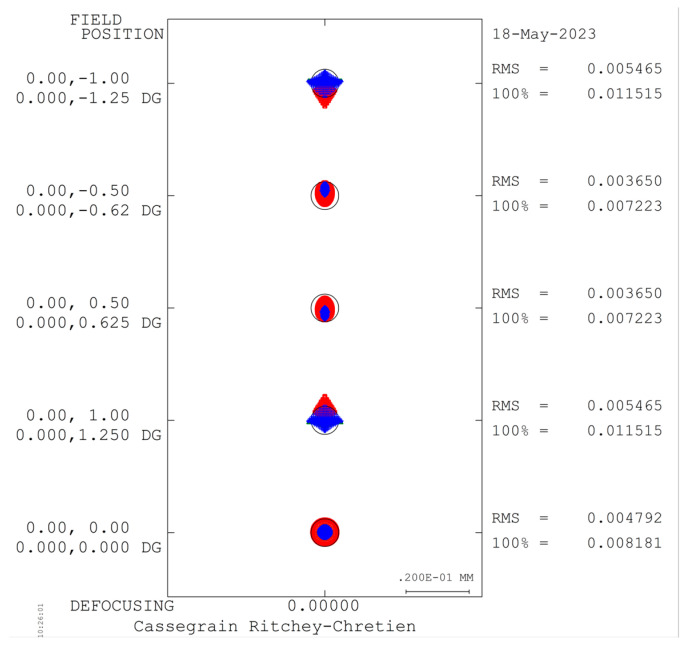
Spot diagram of the forward telescope system.

**Figure 7 sensors-23-05050-f007:**
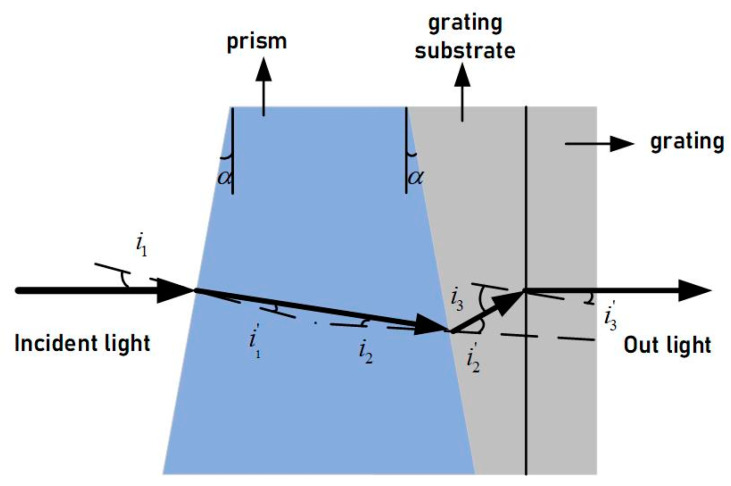
Schematic of PG component.

**Figure 8 sensors-23-05050-f008:**
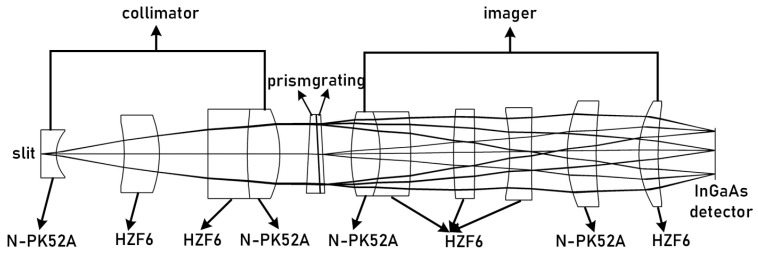
Schematic diagram of the spectroscopic system.

**Figure 9 sensors-23-05050-f009:**
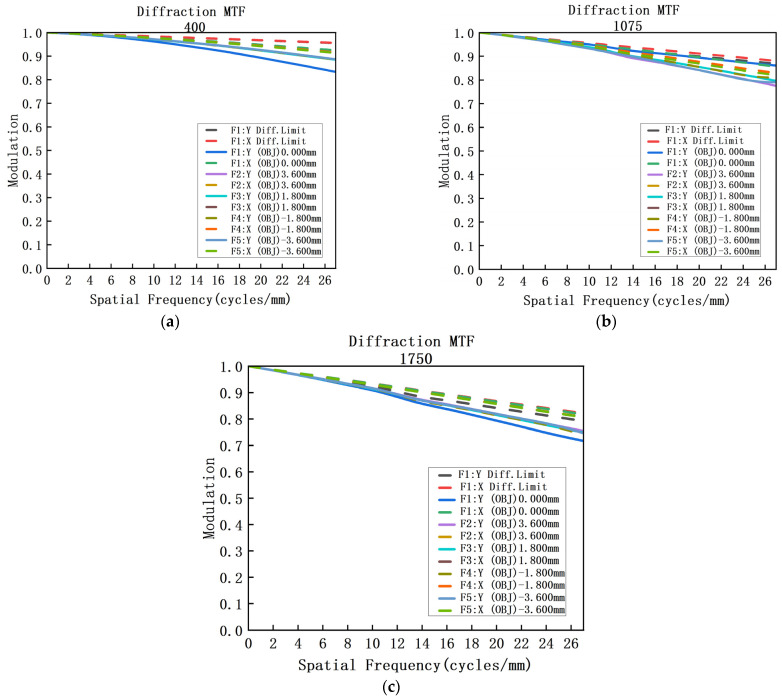
(**a**) MTF diagram of the 400 nm spectroscopic system; (**b**) MTF diagram of the 1075 nm spectroscopic system; (**c**) MTF diagram of the 1750 nm spectroscopic system.

**Figure 10 sensors-23-05050-f010:**
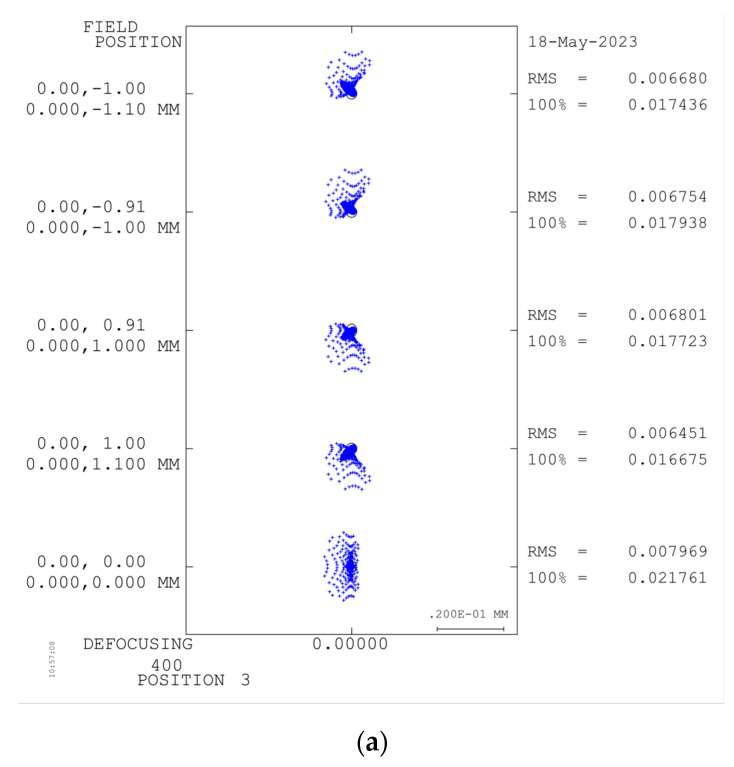
(**a**) 400 nm spectroscopic system spot diagram; (**b**) 1075 nm spectroscopic system spot diagram; (**c**) 1750 nm spectroscopic system spot diagram.

**Figure 11 sensors-23-05050-f011:**
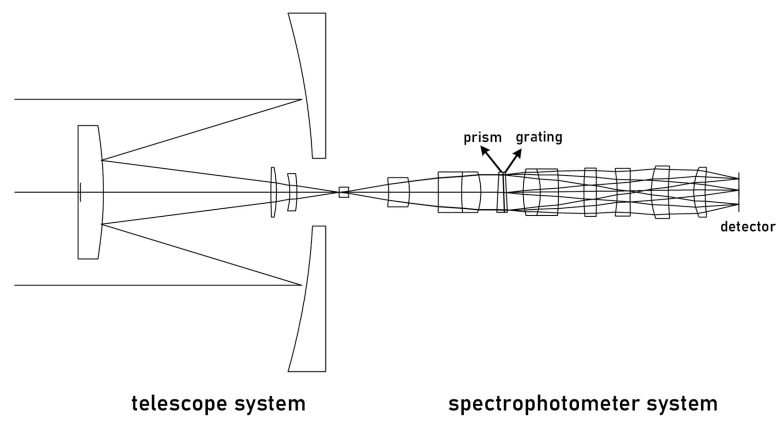
Prism-grating type spectrometer overall optical structure diagram.

**Figure 12 sensors-23-05050-f012:**
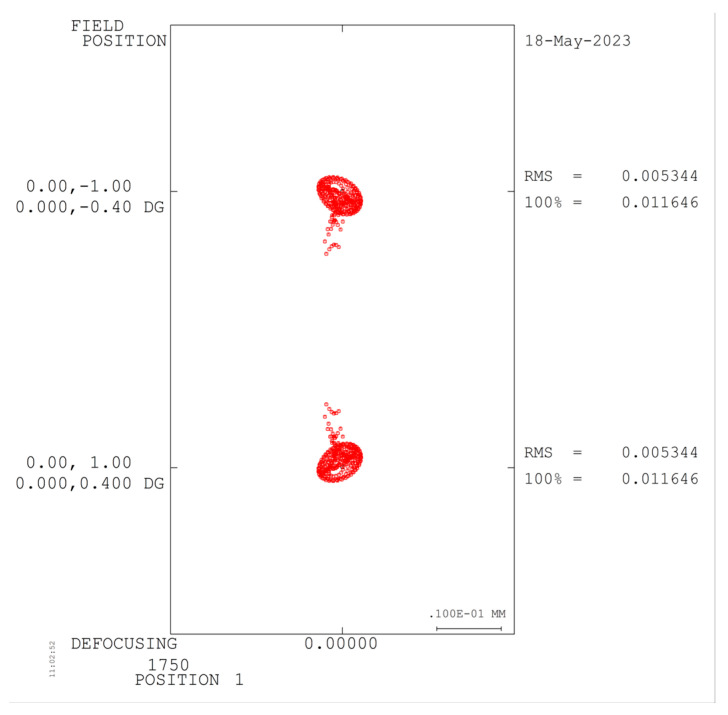
Spot Diagram of the overall optical system.

**Figure 13 sensors-23-05050-f013:**
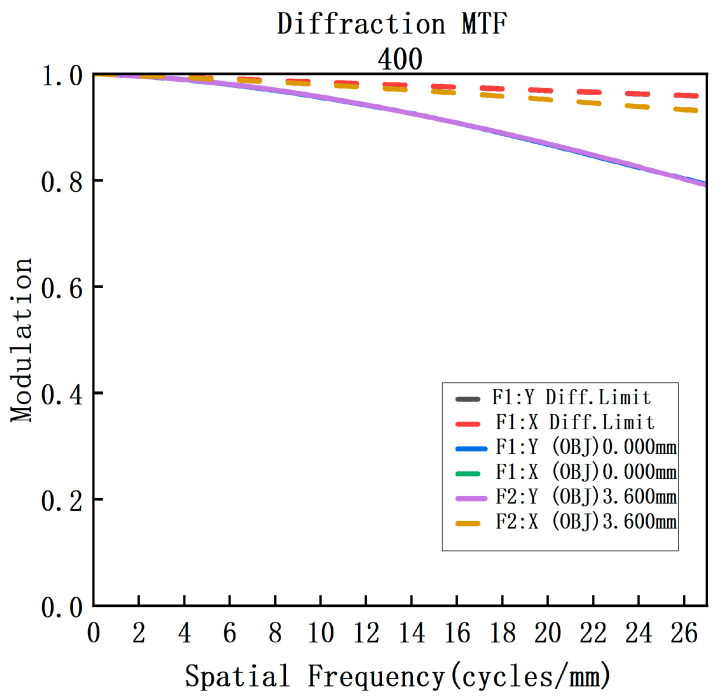
MTF diagram of the overall optical system.

**Table 1 sensors-23-05050-t001:** The Abbe number and relative dispersion parameters table of the selected optical glass are as follows.

Material	$n_{\lambda_{1}}$	$n_{\lambda_{2}}$	$n_{\lambda_{3}}$	υ	p
N-PK52A	1.4418	1.4284	1.4251	21.2345	0.1976
HZF6	1.7403	1.7269	1.7236	7.6521	0.1347
SILICA	1.4701	1.4495	1.4415	15.7283	0.2788
ZNS	2.5589	2.2874	2.2677	4.4198	0.0679
CAF2	1.5077	1.4896	1.4849	21.4741	0.2064
N-BAK4	1.5869	1.5567	1.5485	14.4631	0.2003
N-BAK2	1.5560	1.5290	1.5214	15.2956	0.2206
N-BAK1	1.5902	1.5607	1.5529	15.0568	0.2041
SK5	1.6061	1.5772	1.5685	15.3837	0.2331
FK51	1.4966	1.4795	1.4744	21.6336	0.2261
LAF20	1.7082	1.6663	1.6571	13.0154	0.1821
K10	1.5172	1.4906	1.6677	14.2701	0.2254
P-LAK35	1.7167	1.6781	1.6677	13.8225	0.2102
N-PK52A	1.4418	1.4284	1.4251	21.2345	0.1976

**Table 2 sensors-23-05050-t002:** Data of secondary spectrum for different material combinations.

Material Combination	$\Delta L^{\prime }{}_{\lambda }\;(\mathrm{mm})$
N-PK52A, HZF6	0.68
CAF6, HZF6	0.76
FK51, HZF6	0.96

**Table 3 sensors-23-05050-t003:** System design technical indicators.

Performance	Value
Spectral range/nm	400–1750
F-number	3.34
Field of view/[(°)]	2.5
Slit size/mm	6.45
NA of imaging objective	0.148
System size/mm × mm × mm	145 × 48 × 48
Focal length of telescopic system/mm	147

## Data Availability

The data in this article is confidential and therefore cannot be disclosed.

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
