# Peer review of "Design of a Prism-Grating Wide Spectral Range Transmittance Imaging Spectrometer"

_sensors, 2023, doi:10.3390/s23115050_

Round 1
Reviewer 1 Report
Imaging spectrometers have found a wide range of applications in various fields, including land, sea and atmosphere, which enjoy great significance. The authors introduced a prism-grating type wide-spectrum transmittance imaging spectrometer that corrects for axial chromatic aberration and secondary spectrum, while ensuring the optical axis is perpendicular to the image plane for easy post-installation adjustment. Some analysis and test results are also introduced in detail.
However, there are some problems to be further addressed as well:
1. In line 11, What does “trans” mean?
2. In line 50, Reference 3 cited incorrectly;
3. In line 48,“ However, the typical off-axis structure of imaging spectrometers makes installation and adjustment challenging, and the manufacturing cost of concave and convex gratings is high, limiting their use in civilian markets.” However, the manufacturing cost of prism-grating is also high.
4. Need to specify what type of prism grating is it, Volume Bragg grating or Surface relief grating. In the meantime the authors need to analyse the diffraction efficiency of the prism grating.
Reviewer 2 Report
The paper is (to my understanding) correct and valid, and the English is good and understandable. However the main problem I see with the paper is that the amount of novel information or research results is very small. Almost all the equations used in the text are well known. The optimization presented in the paper is valid, but it is just one example system, optimized with a commercial software, so the general learning is very much limited. The figures are not high quality, mostly taken directly form the software, without commenting them or making them look nice. Some equations are wrong. Also, the logic (arrangement of explanations) is confusing and there is nothing really new. I think this paper does not deserve publication in this form.
Sorry for the harsh feedback. To help the authors for better papers in the future some comments (not all):
Figure 1: The design is not explained in the text or figure. Where does the design come from (reference), what are the design data and material? Please indicate the elements (object, grating, detector) in the figure to help the reader to understand.
Figure 2: There is a typo, as axial is used twice ((b) should be lateral). The picture is very often shown in textbooks, nothing new here.
Text after Figure 2, equations (2) and (3): The coordinate system should be shown in the figures, what is y, what is l?
Equation (6) is wrong, the power of a combined system is the weighted sum of powers (distance between powers is relevant).
Figure 3: the text is too small to be readable.
Table 1: These data can be found in any glass-library.
Table 2: I should be clear in the text which system (I assume Fig. 3) the authors talk about, and which material is in which lens. In fact the system should be explained first?
Figure 3: Once again, the figure caption should describe the components: where is the grating, etc?
Figure 6: Typo “grating” not “Grathing”
Figure 7-10: The Spot and MTF-figures are not very nice, just taken from CodeV. Scaling and line-thickness should be used to make them higher quality.
General it is very confusing to show the final system at the end (Fig10), it should be introduced from beginning.
English is mostly okay, but paper logic is missing
Round 2
Reviewer 1 Report
1. Figure 7: Typo. “grating” not “Grathing”.
2. In Figure 7 the grating is illustrated as a surface relief grating. Therefore, it needs to be modified to a Volume Bragg grating.
Reviewer 2 Report
The paper has been improved and it is (to my understanding) correct and valid. However, the main problem I still see with the paper is that the amount of novel information is very small. Almost all the equations used in the text are well known and the design concept is straight-forward.
If the editors decide to publish, I still see some need for improvement:
The English should be checked again, especially the added text in the new version.
The figures showing Spot and MTF (Fig, 5,6,7,10,11,12,13) are not very nice, just taken from CodeV. Scaling and line-thickness should be used to make them higher quality. Maybe several figures can be combined.
The paper has been improved and it is (to my understanding) correct and valid. However, the main problem I still see with the paper is that the amount of novel information is very small. Almost all the equations used in the text are well known and the design concept is straight-forward.
If the editors decide to publish, I still see some need for improvement:
The English should be checked again, especially the added text in the new version.
The figures showing Spot and MTF (Fig, 5,6,7,10,11,12,13) are not very nice, just taken from CodeV. Scaling and line-thickness should be used to make them higher quality. Maybe several figures can be combined.
